
# Development of a moving point source model for shipping emission dispersion modelling in EPISODE-CityChem v1.3

Kang Pan[1], Mei Qi Lim[1], Markus Kraft[1,2,3], and Epaminondas Mastorakos[1,4]

[1]Cambridge Centre for Advanced Research and Education in Singapore LTD., 1 Create Way, #05-05 Create Tower, Singapore, 138602

[2]School of Chemical Engineering and Biotechnology, University of Cambridge, West Cambridge Site, Philippa Fawcett Drive, Cambridge, CB3 0AS, UK

[3]Department of Chemical and Biomedical Engineering, Nanyang Technological University, 62 Nanyang Drive, Singapore, 637459

[4]Department of Engineering, University of Cambridge, Trumpington Street, Cambridge, CB2 1PZ, UK

**Correspondence:** Kang Pan (kangpan@mie.utoronto.ca), Epaminondas Mastorakos (em257@eng.cam.ac.uk)

**Abstract.** This paper demonstrates the development of a moving point source (MPS) model for simulating the atmospheric dispersion of pollutants emitted from ships under movement. The new model is integrated into the chemistry transport model EPISODE-CityChem v1.3. In the new model, ship parameters, especially speed and direction, are included to simulate the instantaneous ship positions and then the emission dispersion at different simulation time. The new model was then applied to

the shipping emission dispersion modelling. The simulated instantaneous and hourly-averaged emission concentrations were compared to those obtained by the commonly-used line source and fixed point source models. More realistic results were observed by using the new model. Furthermore, the simulated results were compared to the measured values at different locations, and reasonable emission concentrations can be predicted by the moving point source model.

## 1 Introduction

Maritime transport plays an important role in the global transportation for passengers and goods. Compared to other transportation modes, such as road and air transport, maritime transport is considered as the most energy efficient and environment-friendly mode (International Maritime Organization (IMO), 2012). Maritime trade has grown rapidly in past years and is expected to keep an average annual growth rate of 3.5% in future five years (Asariotis et al., 2019), leading to an increase of maritime activities. As a result, pollutant emissions generated from ships keep increasing and hence their impact on human

health and environment in coastal cities and harbours has received attention by researchers (Langella et al., 2016; Tzannatos, 2010; Goldsworthy and Goldsworthy, 2015).

To evaluate the contributions of ship emissions on air quality in coastal areas, atmospheric dispersion modelling of the pollutants, such as $NO_x$, $SO_2$ and Particulate Matter (PM), in a regional or city scale by considering the local meteorological conditions, topographical information, turbulent diffusion, and chemical transformations is a useful approach. Different

dispersion softwares such as a Gaussian model and a Eulerian model have been developed and widely applied in numerical simulations (Milazzo et al., 2017; Gibson et al., 2013; De Nicola et al., 2013; Mallet et al., 2018; Kukkonen et al., 2016). The





most common and simplest one is a Gaussian-based model that assumes the dispersion of air pollutant to follow a Gaussian distribution. Merico et al. (2019) applied a steady-state Gaussian-based model − ADMS-5 − to estimate the dispersion of emissions from ships which are mainly in the hoteling and maneuvering phase in the harbour area of an Italian port city Bari.

The same model was also used by Cesari et al. (2016) for a case study of ship emissions in Brindisi, Italy. Another popular steady-state Gaussian plume model, AERMOD, recommended by United States Environmental Protection Agency (EPA) was also widely used by different groups (Gibson et al., 2013; Cohan et al., 2011). Abrutytė et al. (2014) employed AERMOD to simulate the dispersion of $NO_x$ from ships in Klaipeda port. AERMOD was also used by Fileni et al. (2019) and Cohan et al. (2011) to evaluate the contribution of ships on PM emissions in harbour cities. The Gaussian plume model is able to save a lot

of computational cost, however, it suffers from several limitations, such as assuming a steady-state solution, a spatially uniform meteorology and straight line trajectories (Bluett et al., 2004), that make it not suitable under many conditions for air quality modelling.

In addition to the simple Gaussian models, some advanced dispersion models are developed as well. An example is the un-steady Gaussian puff model named CALPUFF, which is able to simulate the effects of time and space-varying meteorological

conditions on pollutant transport, transformation and removal (Bluett et al., 2004). CALPUFF has been widely used for sim-ulating the dispersion of ship emissions. Jahangiri et al. (2018) applied CALPUFF to simulate the average values of the ship emissions on the port area of Brisbane, Australia for the whole of 2013. Poplawski et al. (2011) and Murena et al. (2018) also employed CALPUFF to evaluate the effects of cruise ships on air quality in the harbours of Victoria, Canada and Naples, Italy respectively. Furthermore, a Lagrangian or Eulerian chemistry transport model (CTM) that can solve the advection-diffusion

equation and the atmospheric chemistry to predict the transport and chemical reactions of emission species also received an increasing attention (Pillai et al., 2012; Gariazzo et al., 2007; Gadhavi et al., 2015). Krysztofiak-Tong et al. (2017) evaluated the air pollution contributed from ships and oil platforms in West Africa by using the Lagrangian model FLEXPART. Shang et al. (2019) and Chen et al. (2018) applied the Eulerian based WRF-CHEM model to simulate the influence of ship emissions on harbour cities in China. Liu et al. (2017) studied the impact of ship emissions on Shanghai urban area by using WRF-CMAQ

model. Huszar et al. (2010) evaluated the effects of ship emissions on $NO_x$ and Ozone ($O_3$) in Eastern Atlantic and Western Europe by using an Eulerian model CAMx. Karl et al. (2020) investigated the particles concentrations in the ship plumes by coupling an aerosol dynamic model MAFOR with a 3D Eulerian chemistry transport model EPISODE-CityChem. In these studies, the chemistry transport models (mainly Eulerian models) have shown the ability of predicting the pollutant concentra-tions at the locations of interest, and hence are a good approach for investigating the environmental impact of ship emissions

in coastal cities.

In pollutant dispersion modelling, it is necessary to include an appropriate assumption or model for treating the emission sources. For a typical setup of shipping emission dispersion simulation, the definition for an emission source usually depends on the ship status, namely that the ship is at berth (hoteling) or on cruise (maneuvering and cruising). Usually, the ship at hoteling phase is treated as a fixed point emission source (Merico et al., 2019; Poplawski et al., 2011; Deniz and Kilic, 2010; Lucialli

et al., 2011; Formentin, 2017), which is a reasonable assumption as the ship stops at the dock and generates emissions from its chimney as a single point. For the ships under movement, different models have been applied to treat the emission sources





in different studies. Iodice et al. (2017) assumed the emissions from the moving ships were generated at multiple fixed points along the predefined navigation route. Saxe and Larsen (2004) used fixed points to represent the average positions of the ships in both maneuvering and hoteling modes. Murena et al. (2018) also treated the ships in maneuvering and navigation modes

as fixed point emission sources. In another group of studies, a line emission source model was widely applied to simulate the moving ships in maneuvering or cruising mode (Poplawski et al., 2011; Kotrikla et al., 2013; Deniz and Kilic, 2010; Lucialli et al., 2011). In the line source (LS) model setup, the ship emission is assumed to be constantly emitted along the entire ship route which is assumed as a straight line. In some other cases, ships in hoteling or maneuvering modes were treated as area sources (Kotrikla et al., 2013; Formentin, 2017; Abrutytė et al., 2014), however, this assumption is not commonly used to treat

ship emissions.

From the above literature review, it is evident that either a (or multiple) fixed point(s) source model or a line source model is commonly used for ships under movement in the air pollution dispersion modelling. However, neither of these assumptions is realistic as the ship position is changing when it is moving. The ship movement is not explicitly included in current air pollution dispersion modelling and leads to a research gap. In this paper, a moving point source (MPS) model was hence developed, that

can update the ship positions at different time based on the ship speed and direction and then simulate the emission release from the moving ships in the dispersion modelling. The new developed MPS model was integrated into the 3D Eulerian chemistry transport model EPISODE-CityChem (Hamer et al., 2020; Karl et al., 2019) and applied to predict the dispersion of $NO_2$ species generated by the ships in cruising mode in a simplified simulation, and the simulated results were compared to those obtained by using LS and fixed point source (FPS) models. In addition, the new MPS model was applied to a real case study

to predict the concentrations of $NO_2$ and $PM_{2.5}$ species contributed by all ships around the Singapore city, and the simulated results were compared to the measured values in different observation stations. The MPS model introduces a new approach for treating the ships and other objects under movement in the atmospheric dispersion modelling, and will increase the knowledge of the atmospheric environment modelers. The model setups and important simulation results are presented in this paper.

## 2    Methods

The MPS model developed in this paper was integrated into the chemistry transport model, EPISODE-CityChem (Hamer et al., 2020; Karl et al., 2019), which is an open source Fortran based code. EPISODE-CityChem is a city-scale chemistry extension of the dispersion model EPISODE, which is originally developed by Slørdal et al. (2003, 2008). In this section, the dispersion model, the MPS model, simulation setups and configurations of the case studies are introduced.

### 2.1    Dispersion model

EPISODE-CityChem simulates the transport, chemical reactions and deposition of pollutant species in both a 3D Eulerian grid and a ground-level sub-grid (Hamer et al., 2020; Karl et al., 2019). A typical Eulerian grid has a horizontal resolution of 1 km by 1 km, while the vertical grid size varies from several meters (near ground) to several hundreds meters (higher layer) with a total height up to several kilometers. The sub-grid has a better resolution with a typical size of 100 m by 100 m horizontally.





The governing advection-diffusion and mass conservation equations for the averaged concentrations in the main Eulerian grid model are indicated as:

$$\frac{\partial C_i}{\partial t} + \frac{\partial(uC_i)}{\partial x} + \frac{\partial(vC_i)}{\partial y} + \frac{\partial(wC_i)}{\partial z} = \frac{\partial}{\partial x}\left(K^{(H)}\frac{\partial C_i}{\partial x}\right) + \frac{\partial}{\partial y}\left(K^{(H)}\frac{\partial C_i}{\partial y}\right) + \frac{\partial}{\partial z}\left(K^{(z)}\frac{\partial C_i}{\partial z}\right) + R_i - S_i \qquad (1)$$

$$\frac{\partial u}{\partial x} + \frac{\partial v}{\partial y} + \frac{\partial w}{\partial z} = 0 \qquad (2)$$

where $C_i$ is the concentration of species $i$ ($i$=1:$N$, where $N$ is the total number of species), $u$, $v$ and $w$ are the three wind velocity components, $K^{(H)}$ and $K^{(z)}$ are the horizontal and vertical eddy diffusivities, and $R_i$ and $S_i$ represent the source and sink terms. The estimation of eddy diffusivities is based on the mixing length theory (K-theory) (Slørdal et al., 2003). In the simulation, the emission source term, wind field and other meteorological conditions are assumed hourly constant. The emission species will be simulated until it is fully diluted or outside of the simulation domain. The photochemistry simulated in the Eulerian grid has several options, such as EMEP45 (Walker et al., 2003), EmChem03mod and EmChem09mod (Simpson et al., 2012), that are modified or updated from the European Monitoring and Evaluation Programme (EMEP) (Simpson, 1995). In this study, the chemical scheme applied is EmChem09mod.

In the sub-grid receptor model, the pollutants generated by emission sources (either a point or a line source) are calculated by using simple Gaussian models. The LS model used in the EPISODE-CityChem package is a steady-state integrated Gaussian plume model (HIWAY-2) with a simplified street canyon model (SSCM), which will affect the concentrations on the receptor points close to the line source (usually within an influence distance of 500 m). The emitted mass from line sources will be integrated into the 3D Eulerian model in each simulation time step. For the point source, a Gaussian segmented plume model, SEGPLU (Walker and Grønskei, 1992), with the use of a Weak-wind Open Road Model (WORM) (Walker, 2011) meteorological pre-processor (WMPP) is implemented to treat the pollutants released from an individual point source as discrete emissions of finite length plume segments, that emitted in each time interval ($\Delta t$). In the calculations, the transport, growth, chemical reaction and deposition of the plumes will be estimated based on the local meteorological conditions where the plumes stay, and the plume mass will be integrated into the Eulerian grids when the segmented plume grows to a predefined size (usually when $\sigma_y/dy$=4 or $\sigma_z/dz$=4, where $\sigma_y$, $\sigma_z$ are Gaussian dispersion length scales in cross wind direction and vertical direction for a plume, and $dy$, $dz$ are the horizontal and vertical sizes of an Eulerian grid cell). The existing plumes will contribute to the concentrations in the sub-grid receptors. The emission concentration at the receptor points will be finally estimated as the sum of the Eulerian grid concentration and contributions from line and point sources, described as Eq. (3).

$$C_{rec}^t = C_m^{t-1} + \sum_{l=1}^{L} C_l^t + \sum_{p=1}^{P} C_p^t \qquad (3)$$

where, $C_{rec}^t$ is the receptor point concentration at time $t$, $C_m^{t-1}$ is the Eulerian grid concentration at previous time step (estimated by Eq. 1), $C_l^t$ and $C_p^t$ are the concentrations contributed from line and point emission sources, $L$ and $P$ are the total numbers of line and point emission sources. In the sub-grid modelling, the stack downwash, dry and wet deposition, and plume rise and penetration are considered as well, and the photochemistry applied is EP-10 plume scheme (Karl et al., 2019). More details about the EPISODE-CityChem software can be found in the papers written by Hamer et al. (2020) and Karl et al. (2019).





## 2.2 The moving point source model

As found from the literature review, LS and FPS models are the common approaches to simulate the emissions generated by the moving ships, however, they are not realistic as they cannot update the instantaneous ship positions. The MPS model is hence developed to fill the gap in the pollutant dispersion modelling.

In the MPS model, five new parameters are defined to determine the ship movement route, as presented in Fig. 1. The most important parameters are ship speed and direction, which can be easily captured from the maritime online databases, such as MarineTraffic. Three additional parameters, namely turning angle, start and stop time, are defined to customize the ship movement, based on more accurate ship travel information. The new variables for each ship are only updated hourly and hence keep constant for each simulation hour. The detailed descriptions about the five model parameters are summarized in Table 1.

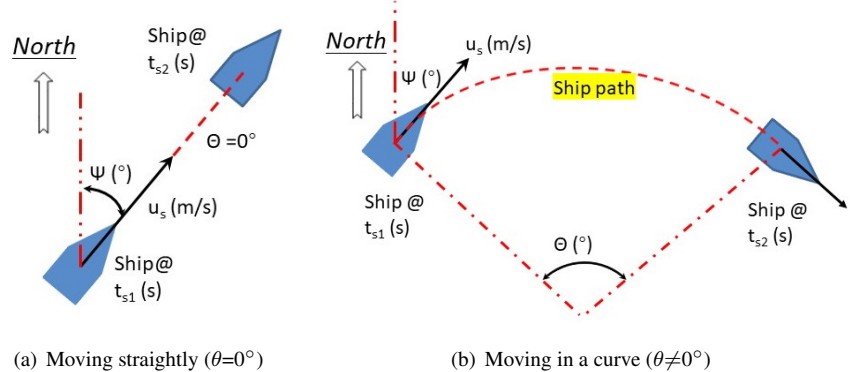

(a) Moving straightly ($\theta=0°$)        (b) Moving in a curve ($\theta\neq0°$)

**Figure 1.** Illustration of the moving point source model. The ship variables in the figures are explained in Table 1.

Based on these five variables, the ship position is estimated and updated in each simulation time step ($\Delta t$). As shown in Fig. 2, ship emission is assumed to be emitted in a virtual point of the short ship line that represents an average ship position for each individual time step. In this model, the ship movement parameters, mainly ship speed, direction and turning angle, are constant for each simulation hour, and hence, the virtual point is usually taken as the middle point of the short travel distance during each $\Delta t$. Apparently, the ship and plume positions will be more realistic when time step is smaller. For the time step

that ship starts or stops to move, the estimation of the virtual point takes time factor into account, as presented in Eqs. (4) and (5).

$$\overrightarrow{x}_i = \overrightarrow{x}_{i-1} + 0.5 d\overrightarrow{x}_i \frac{t_i - t_{s1}}{\Delta t}; \text{(if } t_{i-1} \leq t_{s1} \leq t_i) \tag{4}$$

$$\overrightarrow{x}_i = \overrightarrow{x}_{i-1} + d\overrightarrow{x}_i - 0.5 d\overrightarrow{x}_i \frac{t_{s2} - t_{i-1}}{\Delta t}; \text{(if } t_{i-1} \leq t_{s2} \leq t_i) \tag{5}$$

where, $\overrightarrow{x}_i$ is the virtual position $(x,y)$ of a ship during $i^{th}$ time step in each simulation hour, $d\overrightarrow{x}_i = \overrightarrow{u}_s(t_i - t_{s1})$ (or $\overrightarrow{u}_s(t_{s2} - t_{i-1})$) is the actual ship travel distance in $i^{th}$ time step, $t_i=i\Delta t$ is the actual time difference from the start of each simulation




**Table 1.** Setup of the moving point source model.

| Parameter | Description | Value | Note |
|---|---|---|---|
| speed ($u_s$) | speed at which point source is moving | $\geqslant 0$ m s$^{-1}$ | (1) $u_s = 0$ m s$^{-1}$: fixed point (e.g. ships at berth); |
| | | | (2) $u_s > 0$ m s$^{-1}$: moving point (e.g. ships under cruise). |
| direction ($\Psi$) | direction at which point source is moving | $0 \sim 360°$ | $0°$: north; $90°$: east; $180°$: south; $270°$: west. |
| turning angle ($\theta$) | ship turning angle | $-360 \sim 360°$ | (1) $\theta = 0°$: moving straightly (Fig. 1(a)); |
| | | | (2) $\theta > 0°$: turning clockwisely (Fig. 1(b)); |
| | | | (3) $\theta < 0°$: turning counter-clockwisely (Fig. 1(b)). |
| start time ($t_{s1}$) | time that ship starts to move in each simulation hour | $0 \sim 3600$ s | (1) $0$ s $\leqslant t_{s1} \leqslant 3600$ s if $\theta > 0°$ (when ship is moving straightly, as shown in Fig. 1(a)); |
| | | | (2) in current version, $t_{s1} = 0$ s if $\theta \neq 0°$ (when ship is moving in a curve, as shown in Fig. 1(b)). |
| stop time ($t_{s2}$) | time that ship stops to move in each simulation hour | $0 \sim 3600$ s | (1) $0$ s $\leqslant t_{s2} \leqslant 3600$ s if $\theta > 0°$; |
| | | | (2) in current version, $t_{s2} = 3600$ s if $\theta \neq 0°$; |
| | | | (3) $t_{s1} < t_{s2}$. |

Note: In each simulation hour, all five variables are assumed constant and only need to be updated hourly.

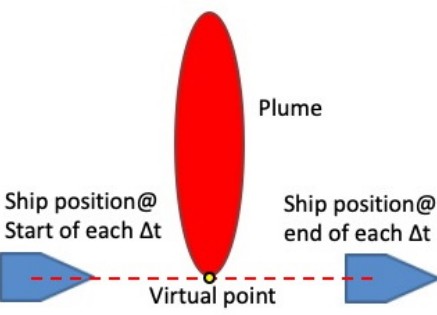

**Figure 2.** Virtual point for plume release in each time step ($\Delta t$).

hour for $i^{th}$ time interval, $\overrightarrow{u}_s$ is the ship velocity. As mentioned above, the five parameters in the simulation are only updated in each hour, assuming that ship is moving in a straight line or a curve with a constant speed during each hour. This is actually a limitation for the current version of MPS model compared to a real ship if its movement parameters change frequently, however, the model used in this study is to address the idea of a MPS model that has the potential of tracking the ship movement and then better simulates the dispersion of ship emissions. The current version of MPS model provides an alternative option for simulating the dispersion of ship emissions in a harbour city. In addition, it should be highlighted that it is possible to define an arbitrary ship movement by using the MPS model, once the real ship movement data collected at different time is added to the MPS input files in the simulation.





## 2.3 Simulation setup

The purpose of this study is to evaluate the new developed MPS model, and two simulations are conducted in Singapore area in this paper. One is a simplified dispersion simulation that only includes moving ships with simplified input conditions, and the results by using the MPS model are compared with the LS and FPS models. Another is a real case study that simulates all the ships around Singapore city by the new model and compared with the emission data measured in different stations.

### 2.3.1 Simplified study

As shown in Fig. 3, the simulation is first conducted to include only a group of moving ships. The simulation domain is set for a 70 km by 70 km area, with a horizontal resolution of 1 km by 1 km in the reference case as suggested by Hamer et al. (2020) and Karl et al. (2019), and the domain has 13 vertical layers, with a height of 10 m in the ground layer and 500 m in the top layer (total height is 3500 m). The sub-grid receptor points (100 m by 100 m horizontal resolution) are created with a receptor height of 1.5 m above ground. In addition, a data point is selected around the coastline, as shown in Fig. 3, to record the emission concentration at every simulation time step.

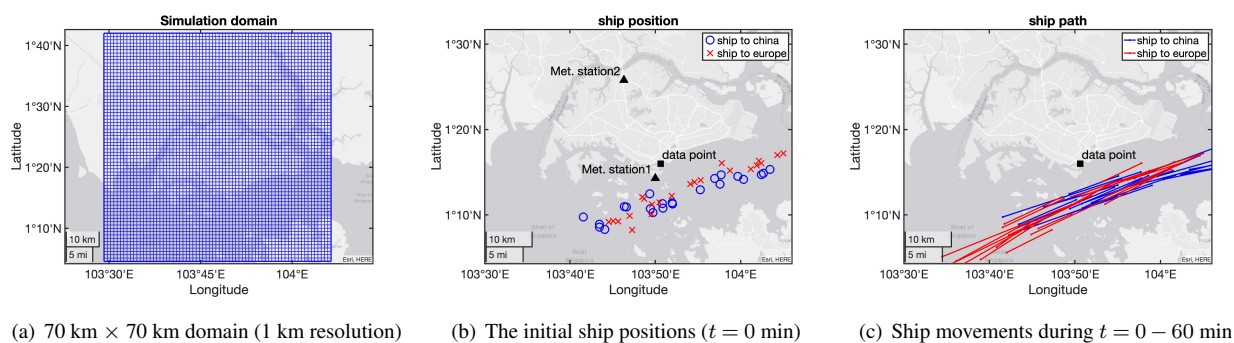

(a) 70 km × 70 km domain (1 km resolution)  (b) The initial ship positions ($t = 0$ min)  (c) Ship movements during $t = 0 - 60$ min

**Figure 3.** Configuration of the simulation domain in Singapore used in the simplified simulation. Symbols: ships to China (Blue circles) and ships to Europe (Red crosses); Lines: ship routes.

To simplify the simulation, a constant meteorological condition taken from two weather stations (as shown in Fig. 3(b)), one in south of Singapore and another in north of Singapore, was applied to the entire simulation period. The details of the weather conditions are shown in Table 2, where the wind inputs in two weather stations were assumed as $2 \, \mathrm{m \, s^{-1}}$ and $180°$ (blown from south to north). A build-in meteorological pre-processor, MCWIND, was used to guess and estimate the local wind speed and direction, based on the input values from the weather stations, and then they were adjusted to the given topography to obtain the 3D divergence-free diagnostic wind field for dispersion modelling (Hamer et al., 2020). Other meteorological parameters (such as vertical temperature gradient) in the simulation area were also estimated by MCWIND. In this paper, the calculated wind field in the ground-level layer for the dispersion modelling is shown in Fig. 4.

In the simplified simulation, a total number of 44 ships with different types and sizes are included. The ships are separated to two groups, where one group (22 ships) is assumed to move towards China and another is heading to Europe. The ship data,





**Table 2.** Meteorological inputs applied in MCWIND pre-processing utility for the simplified simulation.

| $U_{st1}$ (m s$^{-1}$) | $WD_{st1}$ ($^{\circ}$) | $T_{st1}$($^{\circ}$C) | $RH_{st1}$ (%) | $U_{st2}$(m s$^{-1}$) | $WD_{st2}$($^{\circ}$) |
|---|---|---|---|---|---|
| 2.0 | 180 | 32 | 64.3 | 2.0 | 180 |

Note: U: wind speed; WD: wind direction; T: temperature; RH: relative humidity; st1 and st2: weather station 1 and 2.

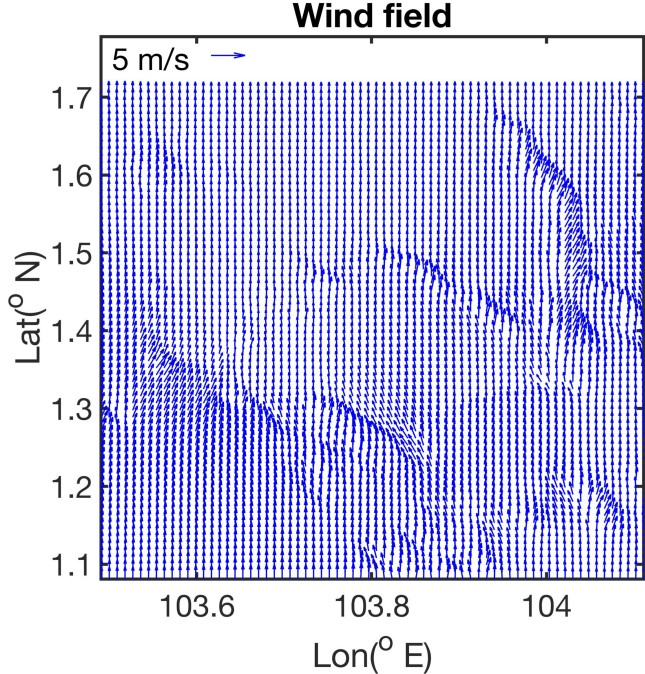

**Figure 4.** The 2D plot of ground-level diagnostic wind field calculated by MCWIND for the simplified dispersion modelling.

such as ship position, speed, direction and gross tonnage, is collected from the online ship resource (such as VesselFinder). In this study, only ships on the China-Europe route (west-east direction) are kept as the initial conditions by removing all other ships (such as those are at berthed or moving in north-south direction), as shown in Fig. 3, and no new ships are included in

the simulation. The dispersion modelling was conducted until all ships moved out of the simulation domain. During the entire simulation, all ships are assumed moving straightly ($\theta$=0$^{\circ}$) and the ship parameters (such as speed and direction) were assumed to be unchanged.

The ships are then divided into different categories (such as liquid bulk ships, dry bulk carriers, container, cargo, etc.) based on those defined in MEET (Methodologies for estimating air pollutant emissions from transport) methodology by Trozzi and

Vaccaro (1999) and Trozzi (2010). The emission rates of main species (such as NO$_x$ and PM) for each ship were then estimated by using the power-based emission factor equation (Eq. (6)) proposed in the MEET method, based on ship's specifications such





as ship type, speed and gross tonnage, as shown in Fig. 5. During the simplified simulation, the emission rates for each ship were constant as the ship operating condition was unchanged, and no background concentrations were used. In addition, the chimney height is assumed to be 30 m for the big size ships (such as the liquid bulk ships) and 10 m for the small ones (such as

the leisure ships), while exit gas is assumed to be at 20 m s$^{-1}$ with 300°C for all ships. In this study, the ship emission sources were treated by using three different models, namely moving point, fixed point and line sources, and the simulated emission profiles were compared. The simulation setups are summarized in Table 3.

$$E_{trip,i,j,k} = \sum_m \left[ t_m \sum_e (P_e \times LF_e \times EF_{e,i,j,k,m}) \right] \tag{6}$$

where, $E_{trip}$ is emission over a trip (kg), $EF$ is emission factor (kg kWh$^{-1}$), $LF$ is engine load factor (%), $P$ is engine

power (kW), $t$ is time (hours), $e$ is engine category (main or auxiliary engine), $i$ is pollutant species (such as NO$_x$, PM), $j$ is engine type (slow, medium and high speed diesel engine, gas turbine and steam turbine), $k$ is fuel type (bunker fuel oil, marine diesel/gas oil, gasoline), $m$ is ship operation mode (cruising, hoteling, maneuvering).

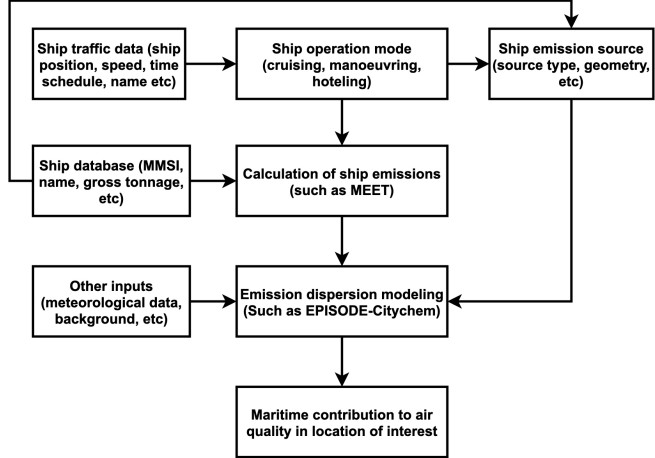

**Figure 5.** Shipping emission dispersion modelling with MEET method (Trozzi, 2010).

### 2.3.2    Real case study

The MPS model was applied to a real case study in this paper as well. The hourly averaged emission values for several

hours (11 am to 4 pm on April 23, 2020) in Singapore were simulated by using the MPS model, and the results at different measurement stations were compared to the measured data. The model setups (such as the grid size) and numerical methods (such as MEET method for emission rate calculation) are same with those used in the simplified simulation, except those (such as the meteorology and background concentrations) introduced in this section. The configuration and setups of the simulation is shown in Fig. 6.





**Table 3.** Setups of shipping emission dispersion modelling.

| Case | Emission source | Time step | Horizontal resolution | Vertical resolution |
|---|---|---|---|---|
| 1 | MPS model | $\Delta t$=15.8 s | dx=dy=1 km (nx=ny=70) | varying dz ($dz_{1-2}$=10 m ... $dz_{13}$=500 m) with total height of 3.5 km |
| 2 | LS model | $\Delta t$=15.8 s | dx=dy=1 km (nx=ny=70) | varying dz ($dz_{1-2}$=10 m ... $dz_{13}$=500 m) with total height of 3.5 km |
| 3 | FPS model | $\Delta t$=15.8 s | dx=dy=1 km (nx=ny=70) | varying dz ($dz_{1-2}$=10 m ... $dz_{13}$=500 m) with total height of 3.5 km |
| 4 | MPS model | $\Delta t$=10 s | dx=dy=1 km (nx=ny=50) | varying dz ($dz_{1-10}$=10 m and $dz_{11-30}$=20 m) with total height of 0.5 km |

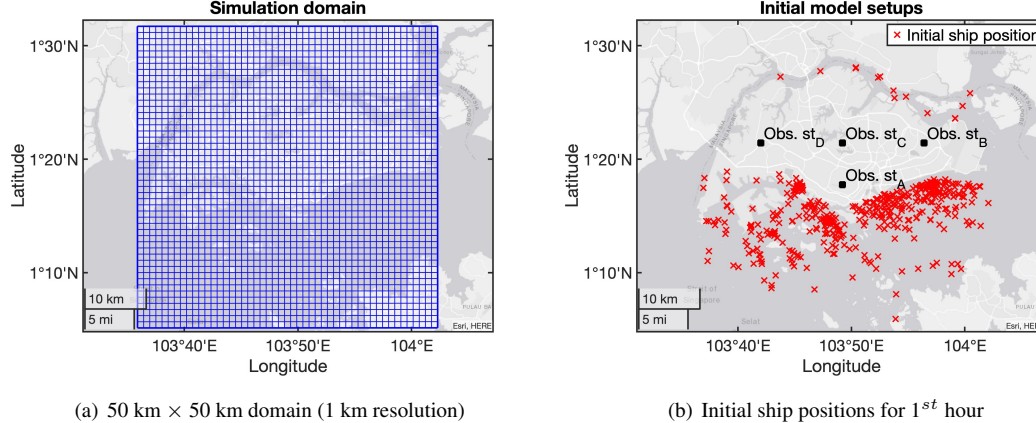

(a) 50 km × 50 km domain (1 km resolution)  (b) Initial ship positions for $1^{st}$ hour

**Figure 6.** Configuration of the simulation domain used in the real case simulation.

The first difference in this real case study compared to the simplified simulation is that all the ships around Singapore city are
included in the simulation, and the ships are only updated in each hour and their emission rates are estimated by using MEET
method (Eq. 6) based on the ship information obtained from the online resource (VesselFinder). Then the meteorological
conditions obtained from Meteorological Service Singapore in each hour are applied to the simulation, while the emission data
obtained from the National Environmental Agency in Singapore was selected as the background concentrations. In this study,
all other model setups and configurations are list as case 4 in Table 3.

## 3 Results and Discussion

In this section, the results for two studies are presented. The first one (section 3.1 and 3.2) is results by comparing the MPS
model with LS and FPS models for a simplified simulation. The air pollution dispersion modelling was first conducted with





only one ship in the simulation domain, and the plume structures simulated by different emission models are compared. The
emission source models were then applied to the additional simulation cases (cases 1-3) that include more ships in China-
Europe direction near Singapore. The instantaneous results and the hourly averaged $NO_2$ values simulated by different emission
models are presented, as both of them are important for evaluating the impact of the pollutant emissions on the locations of
interest. The second part (section 3.3) is a real case study (case 4) that compares the predicted hourly averaged $NO_2$ and $PM_{2.5}$
concentrations by the MPS model at the measurement stations with the measured data.

**3.1  Simplified simulation — preliminary comparisons of different emission source models**

The new MPS model was first tested by simulating only one ship, which moves from east to west side. In this preliminary
simulation, the ship movement parameters are constant, and all other conditions such as wind speed and direction are same as
mentioned in Table 2.

Figure 7(a) presents the instantaneous $NO_2$ concentration near ground simulated by the MPS model. Based on the 2D plots,
it clearly shows that the species concentration inside the plume is gradually reduced in the opposed direction to the ship
movement, which is reasonable. As the ship moves to west-south direction and keeps emitting emissions at different positions
along its route, the early generated emission will be transported by wind to further north and then diluted, and hence, the
emission plume is formed with minimum concentration at the east side and peak value at the west side. The simulated results
indicate that the MPS model gives a quite reasonable prediction for the distribution of emissions released by a moving ship.

In comparison, the LS model gives quite different results, as shown in Fig. 7(b), that the simulated $NO_2$ species is distributed
in a much wider area with a relatively smaller peak concentration. In the dispersion modelling, a line source is a very common
model for treating a moving ship, assuming that the ship continuously generates emissions along the entire line in the simula-
tion. As a result, more emissions appear near the entire ship route and then gradually diluted in the downwind side. Compared
to the real condition, it is unrealistic as the ship keeps moving and is not able to emit emissions from the entire ship route
simultaneously. Furthermore, since the total emission rates (g s$^{-1}$) generated by the ship are same for the MPS and LS models,
the $NO_2$ emission rate at each point along the ship route (or saying emission rate intensity (g s$^{-1}$ m)) in the LS model is much
smaller than the MPS model. Hence, the maximum $NO_2$ concentration generated by the LS model has a relatively smaller
value than the MPS model.

In addition, the simulated emission profiles by using a FPS model are illustrated in Fig. 7(c). The FPS model is another
commonly used assumption for treating the moving ship in the literature. In this study, the moving ship is assumed to stay in
the middle point of the ship routine in each hour. As shown in Fig. 7(c), the $NO_2$ emission is blown to north by wind from the
ship point and then diluted. Since the ship position is assumed unchanged during each hour in the simulation, the emission is
distributed in a much smaller area with a much larger concentration compared to the other two models. Clearly, the FPS model
cannot reveal the effects of ship movement on emission dispersion.



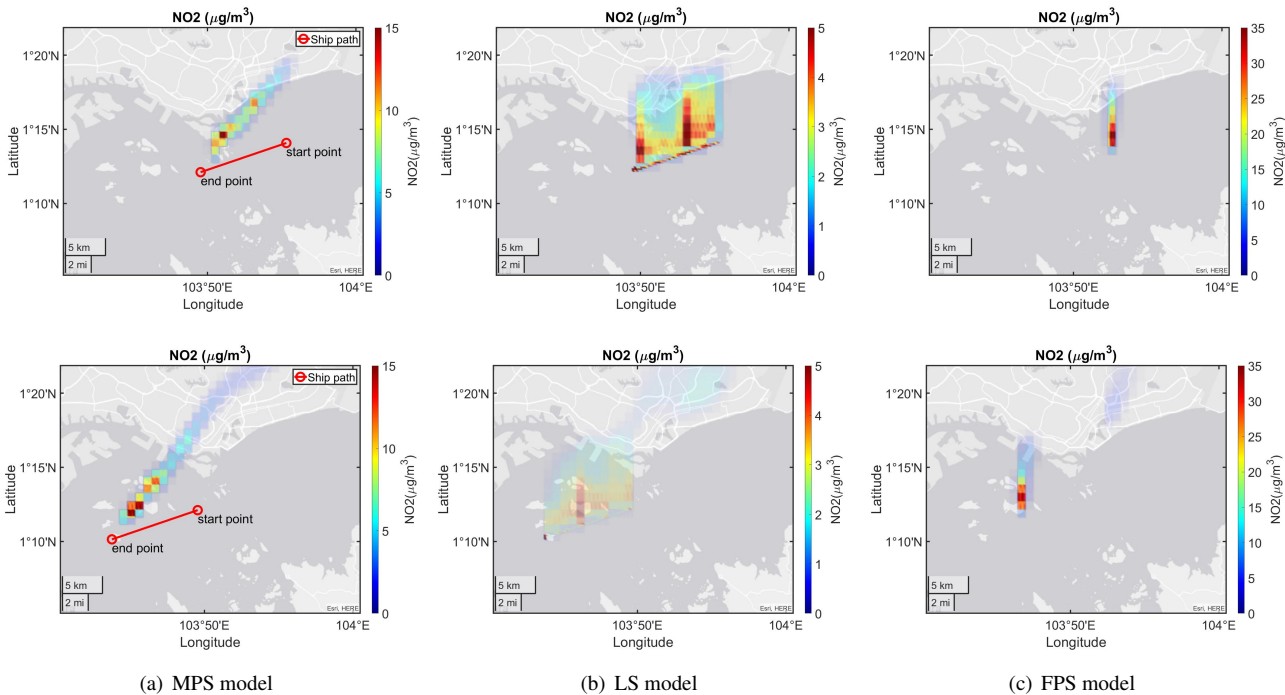

(a) MPS model       (b) LS model       (c) FPS model

**Figure 7.** Instantaneous $NO_2$ concentrations near ground generated by one moving ship. $1^{st}$ row: At $t = 60$ min; $2^{nd}$ row: At $t = 120$ min.

## 3.2 Simplified simulation − results for case studies with more ships

After comparing the three emission source models for only one ship simulation, the three models were applied to a simplified study (cases 1-3) with 44 ships involved, in order to further evaluate the performance of different models for predicting the effects of moving ships on air quality in coastal cities. Both of the instantaneous and average results are presented in this section to fully compare the different emission models. The meteorological conditions and simulation setups are same as presented in Tables 2 and 3.

### 3.2.1 Simulated results by using the moving point source model

In the case study for more moving ships, the simulation was first conducted by using the MPS model (case 1), and the instantaneous ground-level $NO_2$ concentrations at different time around Singapore area are plotted in Fig. 8. Based on the 2D plots, it shows that the $NO_2$ emission moves to north from the ship positions and forms the higher concentration at $t = 60$ min compared to other simulation time, as most of ships are passing the same area during the first 60 minutes (Fig. 3(c)). The gas species then moves to west and east directions as the two groups of ships move towards their destinations, and the gas concentration is continuously diluted in the following simulations as the ships keep moving out of Singapore area.



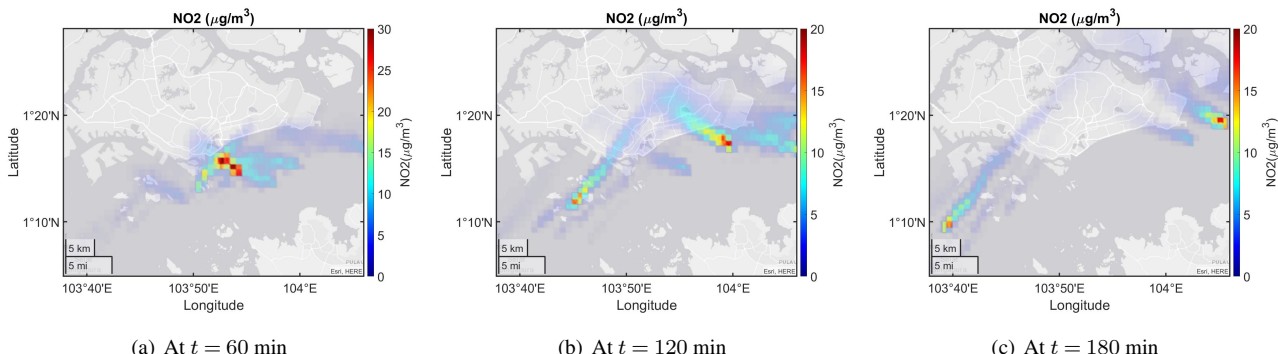

(a) At $t = 60$ min        (b) At $t = 120$ min        (c) At $t = 180$ min

**Figure 8.** Instantaneous NO$_2$ concentrations near ground by using MPS model (case 1).

Figure9 illustrates three vertical NO$_2$ concentration profiles (west-east vertical plane) at $t = 60$ min. From these figures, it shows that less NO$_2$ species arrives at the ground when the plumes are closer to ships (Fig. 9(a)), and then the gas species will

be transported vertically to the ground as the plumes move to the downwind direction (Fig. 9(b)). This is mainly attributed to the plume rise effects that the gas species exits the ship chimney with a certain velocity (in the simplified simulation, the exit velocity is assumed as 20 m s$^{-1}$ for all 44 ships), and then the gas species will be blown by the wind (south to north) and only reaches the ground at a certain distance in the downwind direction. As a result, the peak NO$_2$ concentrations at ground level appear on the locations that are far away from the ship routes but not near the ships, as shown in Fig. 8. As the emissions move

further in the downwind direction, the plumes will be diluted vertically until fully disappeared, as shown in Fig. 9(c).

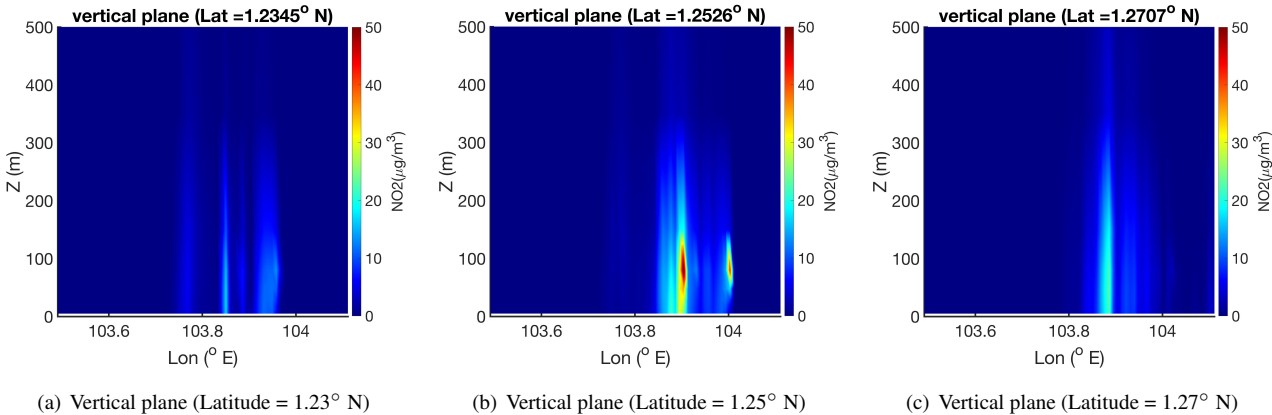

(a) Vertical plane (Latitude = 1.23° N)        (b) Vertical plane (Latitude = 1.25° N)        (c) Vertical plane (Latitude = 1.27° N)

**Figure 9.** Vertical NO$_2$ profiles (west-east direction) at $t = 60$ min.

In addition, the time history of NO$_2$ concentration recorded at the data point (shown in Fig. 3(b)) is also plotted as shown in Fig. 10, where it indicates that there are two peaks for NO$_2$ concentration when using the MPS model. The time history is reasonable. Based on the NO$_2$ curves, it indicates that the emission species generated by the ships take around 30 minutes to



reach the data point, and hence, the two peaks should be induced by the transport and accumulation of emissions generated
during the first 60 minutes. As shown in Fig. 11, a large group of ships pass by or are close to the data point during the first 30
minutes and lead to a continuous emission accumulation to form the first peak concentration, and another group of ships pass
by the data point later (from 40 to 60 minutes) to generate the second peak value. After 60 minutes, most of the ships have
passed the data point (Fig. 12), and hence the NO$_2$ concentration is continuously decreased. The time series of 2D plots in Fig.
8 and the NO$_2$ concentration curve in Fig. 10 reveal that the effects of ship movements on emission distributions can be well
270   captured by using the MPS model.

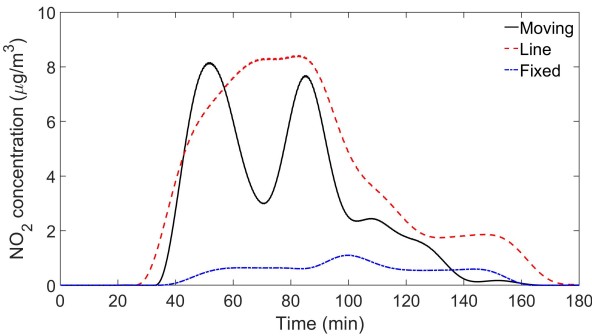

**Figure 10.** Time history of NO$_2$ concentration at data point simulated by using different emission source setups.

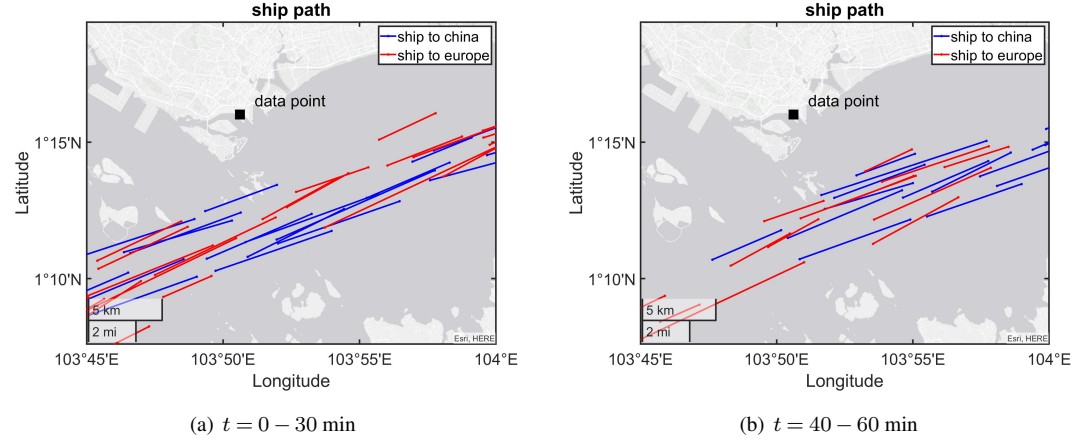

(a) $t = 0 - 30$ min                    (b) $t = 40 - 60$ min

**Figure 11.** Ship movements during $t = 0 - 60$ min.

### 3.2.2   Comparison of three emission source models — instantaneous value

In the simplified case study, the simulation was then conducted by treating each moving ship as a line source (case 2). The
instantaneous NO$_2$ concentrations contributed from the two groups of ships are plotted in Fig. 13 for different simulation time.





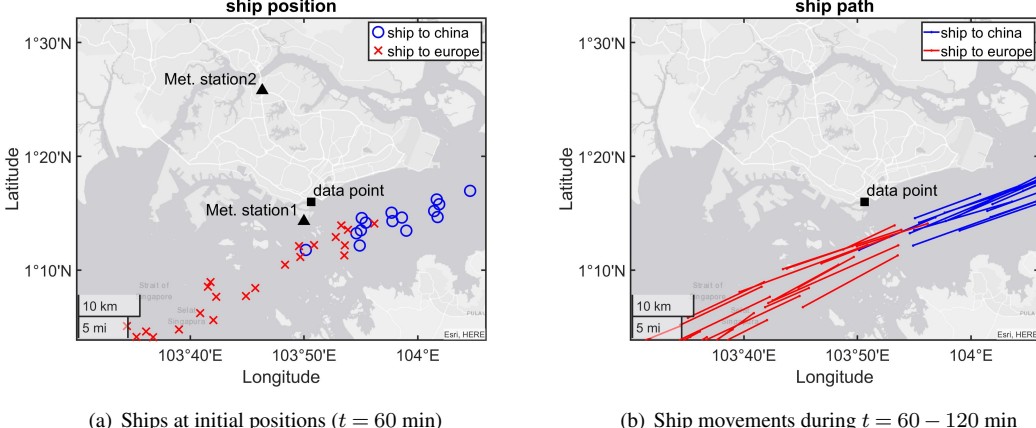

(a) Ships at initial positions ($t = 60$ min)  (b) Ship movements during $t = 60 - 120$ min

**Figure 12.** The ship initial positions and movements during $t = 60 - 120$ min.

Compared to the MPS results (Fig. 8), it clearly shows a much wider $NO_2$ distribution in Singapore area when using the LS

275    model to simulate the moving ships, due to the continuous emission generation along the entire ship routes. For the LS model,

the generated emissions have the continuous impact on a specific area, while the emissions emitted by the MPS model only

have transient impact on the same area. As a result, when ships are concentrated in a small region (as shown in Fig. 3), the

integration of simulated $NO_2$ emission generated by line sources induces a higher peak concentration than the MPS model

(Fig. 13(a)), although the emission rate intensity for each line source is smaller as mentioned in section 3.1. As expected, when

280    the ships are separated, the maximum $NO_2$ concentration for the line source becomes smaller than the MPS model, as shown

in Figs. 13(b) and 13(c).

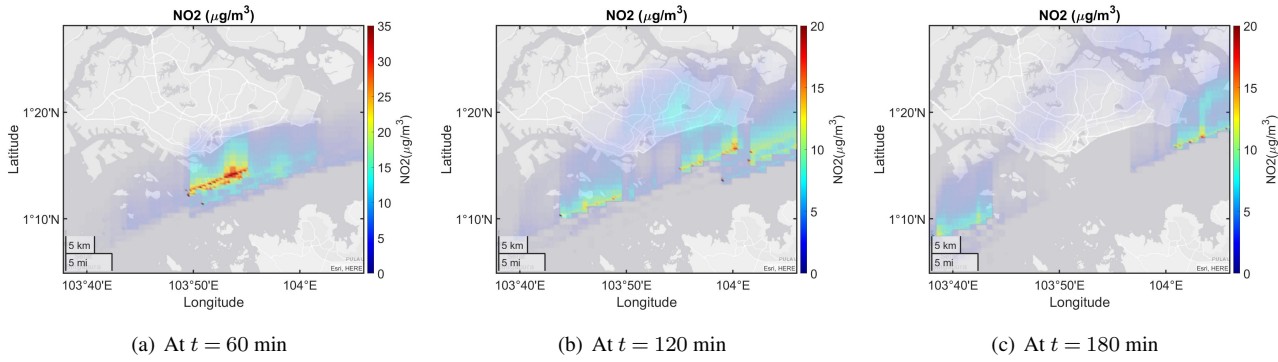

(a) At $t = 60$ min  (b) At $t = 120$ min  (c) At $t = 180$ min

**Figure 13.** Instantaneous $NO_2$ concentrations near ground by using the LS model (case 2).

The $NO_2$ time history curve for the LS simulation is also obtained as shown in Fig. 10. Compared to the MPS model, this

simulated $NO_2$ concentration reaches to its peak at around $t = 65$ min and then is kept for around 15 minutes before it drops.

In EPISODE-CityChem, hourly based simulations are conducted, and all the conditions such as meteorological parameters





and emission setups are constant for every 60 minutes' simulation. As shown in Figs. 3 and 12, more ships pass the data point during the first 60 minutes and less ships pass by during the $2^{nd}$ simulation period ($t = 60 - 120$ min). When the LS model is used, a constant total $NO_2$ emission rate is generated during the $1^{st}$ simulation period ($t = 0 - 60$ min) and continuously affects the data point, leading to a concentration rise in the $NO_2$ curve to the peak value at around $t = 65$ min (Fig. 10). Then the emission generation and dilution reach an equilibrium condition to maintain a constant peak concentration for a while, until the emissions generated by the ships in the $2^{nd}$ simulation period arrive at the data point. A smaller total emission rate is generated by the smaller amount of ships (during $t = 60 - 120$ min) near the data point area , and hence, the local concentration at the data point is reduced, shown as the $NO_2$ curve in Fig. 10. Clearly, the $NO_2$ concentration history obtained by the LS model cannot reveal the effects of real ship movements on emission dispersion, and hence, it is not an appropriate assumption for simulating the instantaneous emission dispersion for ships in cruising mode compared to the MPS model.

In addition, the simulation was conducted by using the FPS model as well, assuming that the ships are staying at the middle points of the ship routes in each hour. The ground-level $NO_2$ distribution profiles are presented as Fig. 14. As expected, the emissions are distributed as separated plume segments, which are clearly not accurate. In Fig. 10, the $NO_2$ history curve for the fixed point assumption has the smallest peak value, as less $NO_2$ emission can be blown to the location of the data point. The individual plume segments and the smallest single-peak $NO_2$ time history indicate that the FPS model is an inaccurate approach for simulating the emission release and dispersion from the moving ships. Based on the comparisons of the simulation results by using three different emission models, it suggests that the new developed MPS model can simulate more realistic ship movement and then instantaneous emission concentrations generated by the moving ships.

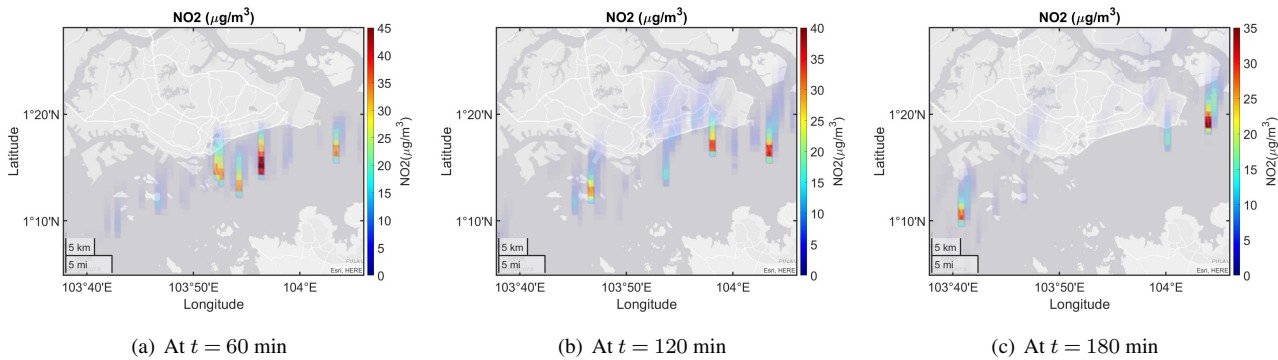

(a) At $t = 60$ min   (b) At $t = 120$ min   (c) At $t = 180$ min

**Figure 14.** Instantaneous $NO_2$ concentrations near ground by using the FPS model (case 3).

### 3.2.3 Comparison of three emission source models − average value

The simulation results in previous sections are the instantaneous $NO_2$ concentrations. In emission dispersion modelling, the average results (usually hourly based) are also important as they can be used for policy decision and for evaluating the long-term environmental impact. In this section, the hourly averaged results by using three different emission source models are compared as well, and the average $NO_2$ concentrations near ground at different simulation time are presented in Fig. 15. Based





on the 2D plots in Fig. 15, it shows that the average NO$_2$ profiles by using the FPS model are much different from the other two setups. As discussed above, the FPS setup is clearly inappropriate for modelling the moving ships.



(a) At $t = 60$ min    (b) At $t = 120$ min    (c) At $t = 180$ min

**Figure 15.** Hourly averaged NO$_2$ concentrations near ground by using different emission models. $1^{st}$ row: MPS model (case 1); $2^{nd}$ row: LS model (case 2); $3^{rd}$ row: FPS model (case 3).

The hourly averaged 2D plots in Fig. 15 also indicate that the simulated NO$_2$ emissions by using the MPS and LS models are distributed in similar area. This is because that the emissions for each ship are emitted along the same ship route for two models, although the location of NO$_2$ species generated by a MPS model changes along the ship route while the LS model emits emission along the entire route continuously. As a result, the accumulated NO$_2$ emissions will cover the similar area for the two model setups and then generate similar results in the hourly averaged evaluations. However, the details of the 315    NO$_2$ distributions (such as the peak concentration locations and values) are different for the two emission models, due to their




natures of treating the emission generation differently in the dispersion modelling. As shown in Fig. 15, the LS model may overestimate the average $NO_2$ concentrations in some locations, compared to the MPS model.

The hourly averaged $NO_2$ concentrations at the data point for three emission source models are presented in Fig. 16. The concentration curves again indicate that the MPS and LS models predict comparable average $NO_2$ concentrations, while the FPS

model gives a much different result. The simulation results suggest that the MPS model should be able to provide an alternative option to predict the hourly averaged emission concentrations and distributions in the air pollution dispersion modelling.

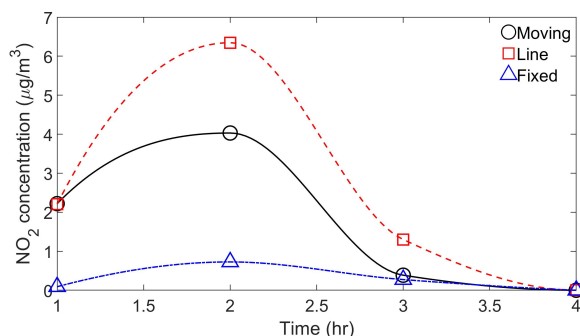

**Figure 16.** Hourly averaged $NO_2$ concentrations at data point simulated by using different emission source setups.

## 3.3 Real case study — comparison with measurement

After comparing with the LS and FPS models in a simplified study, the new developed MPS model was applied to the real case by predicting the emission results generated by all ships (including those under cruise and at berth) around Singapore area

during a couple of hours. The predicted hourly averaged $NO_2$ and $PM_{2.5}$ concentrations are compared to the observed results obtained from the Singapore National Environment Agency online data resource.

Figure 17 compares the concentrations of $NO_2$ and $PM_{2.5}$ predicted by using the MPS model with the values obtained from different stations. Based on the figures, it shows that the new developed MPS model can reasonably predict the emission values at the four stations compared to the measured results, although there are still gaps between simulation and measurement. The

differences may contributed from following aspects. First of all, only the emissions generated from ships around Singapore were included in the simulation, however, in the real world, the emissions measured in the observation stations should be the results contributed from ships and other sources, such as cars and powerplants. In addition, some assumptions were made in the simulation to simplify the model inputs that could induce different results from the real conditions. In the simulation, the meteorological conditions (such as wind speed and direction) are assumed to be hourly constant although a space-varying wind

field is estimated based on the input values at multiple weather stations, while the real wind and temperature are time and space-varying that could highly affect the dispersion of the emissions generated from ships. A constant background concentration was applied for the simulation while the actual value changes in different locations at different time. For the MPS model, each ship is assumed to move at constant speed and direction in each simulation hour and no new ships are included until next hour,





however, in the real world, ships' speed and direction change very frequently and ships can travel to the selected region at any

time. The emission inventory or emission rate calculated by using the MEET method is based on the empirical equations fitted

by the emission data obtained from a ship database, that includes ships with different types and sizes operating under different

conditions (such as cruising and hoteling), and the estimated emission rates may be quite different from the actual values.

Finally, the computational methods and model setups used in the simulation may not reveal the real process. The simulation is

applied to a city-scale region with relatively coarse mesh setup and the emission details at specific locations may not be well

captured, and the local emission distribution may vary highly due to the effects of different factors, such as building effect and

different surface roughness. The chemical reactions of emission species are very complicated and may not be well predicted

by the chemical mechanism applied in the simulation, while the physical changes of the aerosol particles are not simulated by

the model.

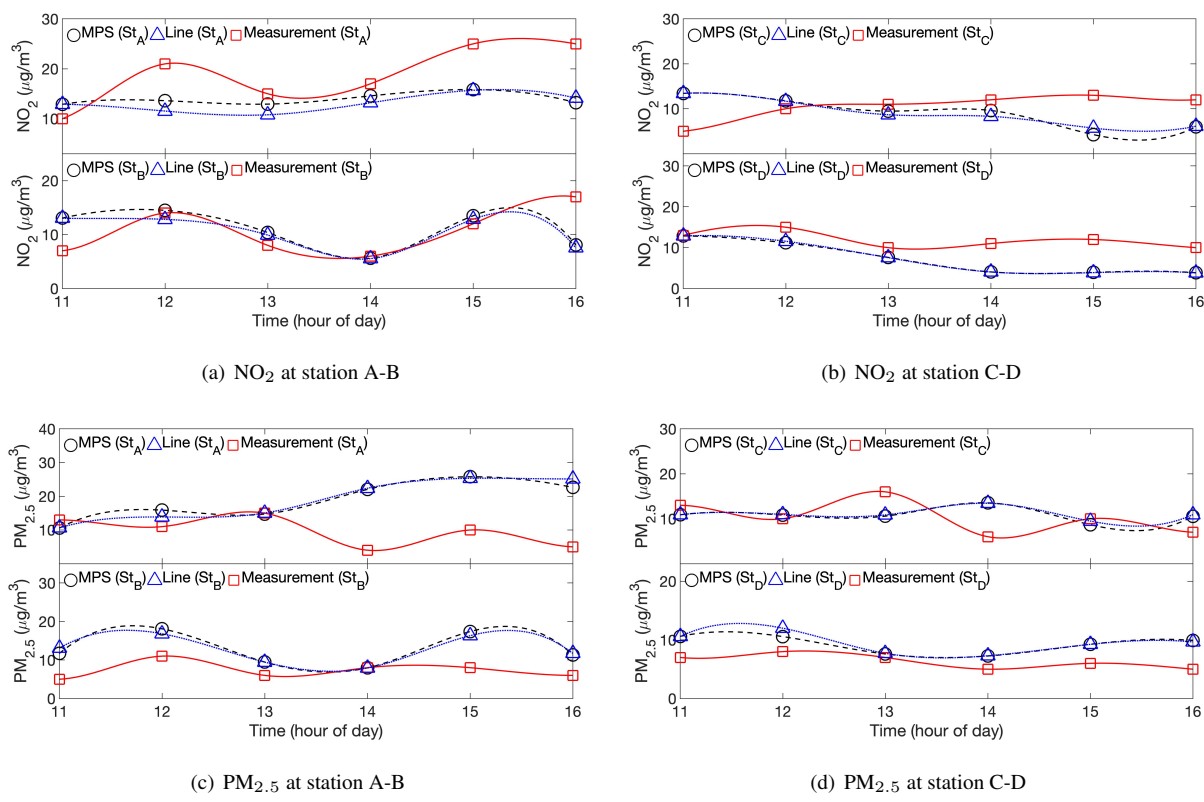

(a) $NO_2$ at station A-B

(b) $NO_2$ at station C-D

(c) $PM_{2.5}$ at station A-B

(d) $PM_{2.5}$ at station C-D

**Figure 17.** Comparison of hourly averaged emission concentrations between simulation and measurement at different locations.

The simulation results by using the LS model are also presented in Fig.17. Based on the figures, it indicates that the $NO_2$

species predicted by the MPS model is slightly better than those predicted by the LS model, especially near observation station

A and C, compared to the measured data. The different results for the two models are mainly contributed by the different

treatments for the moving ships, and the better results may suggest that the new MPS model could capture the impact of ship





movement on the dispersion of emission species and thereby induce a more realistic prediction. At the same time, the simulated PM$_{2.5}$ concentrations by using the two different source models are quite similar. Although it is hard to say that the simulated

results by using the new developed MPS model can predict better than the LS model under all conditions, these comparisons suggest that the MPS model could provide an alternative for the environmental researchers to evaluate the dispersion of the emission species generated from the ships, especially those under cruise. Based on the results in this section and the simplified study, it reveals that the LS model would be a good option to simulate the emission dispersion when a large number of moving ships are in the location of interest, however, the new developed MPS model should be more accurate and could provide more

granularity in the prediction.

## 4   Conclusions

In this paper, a MPS model was developed to simulate the emission generation and transport from the moving ships in pollutant dispersion simulations. For the dispersion modelling, the common assumption is to use a LS or a FPS model to treat the emissions generated by the moving ships. Both models cannot update the ship movements within a certain time period (usually

an hour), and will result in unrealistic emission distribution. In the MPS model, the ship movement parameters, including speed and direction, are used to update the ship positions and then to estimate the emission dispersion at different simulation time. The new developed model has been integrated into the city-scale chemistry transport model, EPISODE-CityChem, and then was evaluated by simulating the atmospheric dispersion of emission species emitted by the ships in Singapore area.

The computational results by using the MPS model were first compared to those obtained from a LS model and a FPS model

in simplified simulations. The results indicated that the new developed MPS model can simulate the ship movement, and hence, predicts more realistic instantaneous concentration profiles for the emission species (such as NO$_2$) generated by the moving ships. In comparison, the LS model will assume a continuous and constant emission rate along the entire ship route, and then results in much different emission profiles and cannot reveal the instantaneous impact of ship movements on air quality in the coastal area. For the FPS model, separated plume segments were observed in the simulation. Clearly, it is unrealistic as the

emission is continuously generated by the moving ships from different positions at different time, and a continuous emission distribution should be formed. The hourly averaged values were compared as well for all three models. The comparison shows that the averaged concentration profiles are similar but with local differences for the MPS and LS models, mainly caused by the different treatments for emission release by the two models although the positions of emission release cover the same ship routes. The FPS model again was proven to be an inappropriate assumption for treating the moving emission source.

In addition, a real case study was conducted as well to further evaluate the MPS model by simulating all ships around Singapore area. Compared to the measured emission data, the MPS model was found that it can reasonably predict the emission concentrations at different measurement stations located in Singapore, although gaps still exist due to the different setups and configurations between simulations and measurements. The LS model was compared in the study as well, and a slightly better result was found for the MPS model at some stations. The real case study together with the simplified study suggest that the

MPS model could be more accurate and provide more details in shipping emission dispersion modelling, compared to the LS





and FPS models, and therefore, the MPS model could be an alternative for the environmental society to evaluate the pollutant dispersion contributed from the moving ships.

*Code availability.* The source code of the MPS model is available at http://doi.org/10.5281/zenodo.4650482 (Pan 2021). The code is written in Fortran 90 and is integrated with EPISODE-CityChem v1.3. The source codes of the EPISODE-CityChem v1.3 and the preprocessing
utilities are accessible in release under the RPL license at https://doi.org/10.5281/zenodo.3549415 (Karl and Ramacher, 2019)

*Data availability.* The following datasets are available upon request from the authors. 1, input and output data of EPISODE-CityChem simulations for simplified cases with 1 ship in Singapore ($\sim$0.6 GB); 2, input and output data of EPISODE-CityChem simulations for simplified cases with 44 ships in Singapore ($\sim$1.1 GB); 3. input and output data of EPISODE-CityChem simulations for the real case study in Singapore ($\sim$0.5 GB).

**Appendix A: Parameter study — time step**

To further evaluate the MPS model, time step is investigated. In the reference case (case 1), the calculated time step is 15.8 s, and in this parameter study, time step is adjusted to two different values of 10 s (case S1) and 30 s (case S2) as shown in Table A1. All other conditions and model setups for all three cases are same.

**Table A1.** Setups of pollutant dispersion modelling for parameter study.

| Case | Emission source | Time step | Horizontal resolution | Vertical resolution |
|------|-----------------|-----------|-----------------------|---------------------|
| S1 | MPS model | $\Delta t$=10 s | dx=dy=1 km (nx=ny=70) | varying dz (dz$_{1-2}$=10 m ... dz$_{13}$=500 m) with total height of 3.5 km |
| S2 | MPS model | $\Delta t$=30 s | dx=dy=1 km (nx=ny=70) | varying dz (dz$_{1-2}$=10 m ... dz$_{13}$=500 m) with total height of 3.5 km |
| S3 | MPS model | $\Delta t$=15.8 s | dx=dy=0.7 km (nx=ny=100) | varying dz (dz$_{1-2}$=10 m ... dz$_{13}$=500 m) with total height of 3.5 km |
| S4 | MPS model | $\Delta t$=15.8 s | dx=dy=1.4 km (nx=ny=50) | varying dz (dz$_{1-2}$=10 m ... dz$_{13}$=500 m) with total height of 3.5 km |
| S5 | MPS model | $\Delta t$=15.8 s | dx=dy=1 km (nx=ny=70) | smaller dz (dz$_{1-4}$=10 m ... dz$_{20}$=500 m) with total height of 3.5 km |

The instantaneous NO$_2$ profiles at ground level for two additional time step simulations are plotted in Fig. A1. Compared
to the reference case (Fig. 8), it indicates that the NO$_2$ profiles at different simulation time are almost same for all three cases, although the local emission distributions and concentrations are slightly different.





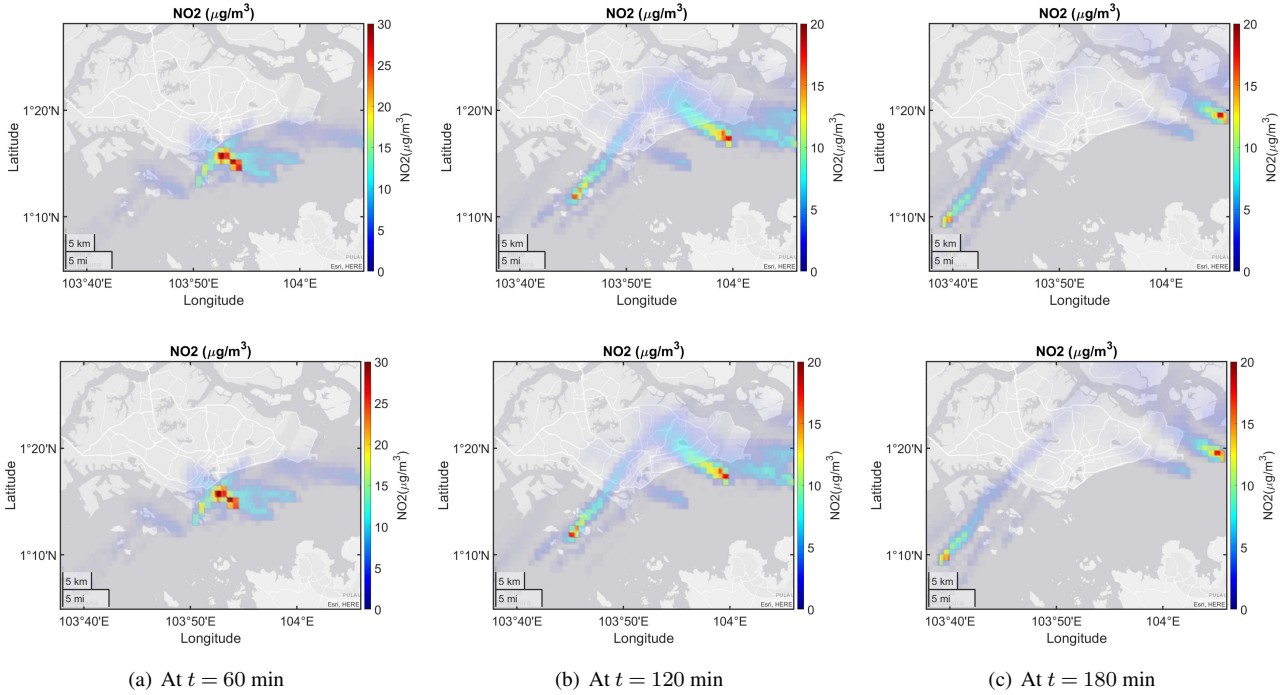

(a) At $t = 60$ min  (b) At $t = 120$ min  (c) At $t = 180$ min

**Figure A1.** Instantaneous NO$_2$ concentrations near ground by using different time steps. $1^{st}$ row: $\Delta t$=10 s (case S1); $2^{nd}$ row: $\Delta t$=30 s (case S2).

In EPISODE-CityChem, parallel simulations of emission dispersion are conducted, that one Eulerian main grid (where dx=dy=1 km) is built up to model the time-dependent advection and diffusion of emission species in the 3D space. At the same time, the emissions emitted from each point source in the sub-grid modelling are treated as finite Gaussian plume segments generated in each time step. The plume size and movement (speed and direction) are estimated based on the local meteorological conditions (mainly temperature, wind speed and direction), in which Eulerian grid cell that the plume start point stays. In next time step, the plume position is updated and then its size and movement parameters are re-calculated based on the new meteorological conditions of the main grid cell, where the plume segment is transported. In addition, when the length scale of the segmented plume ($\sigma_y$ or $\sigma_z$, which is highly affected by the meteorological conditions such as wind speed and temperature) reaches to a pre-defined value (usually 1/4 of the Eulerian grid size), the plume mass will be integrated into the main Eulerian grid cell that the segmented plume locates and then deleted from the sub-grid model.

As the wind field (speed and direction) estimated by EPISODE-CityChem is spatially different (as shown in Fig. 4), the mass and number of plume segments and the values of other parameters (position, size, speed and direction) for each plume segment estimated in the dispersion modelling are different when using different time steps, and hence, the plume prediction in the sub-grid modelling will be different to result in different emission concentrations. However, as the time step reduces to a relatively small value, the impacts of time step on simulation results are negligible as shown in this paper (Figs. 8 and A1).





## Appendix B: Parameter study − grid resolution

Another parameter study was conducted by changing the horizontal (case S3 and S4) and vertical (case S5) grid size. In this section, the dispersion of ship emissions was first simulated by using two different main grid resolutions, where the horizontal

grid sizes (dx=dy) are changed to 700 m (case S3) and 1400 m (case S4), compared to the reference case (case 1 where dx=dy=1 km). To keep the same simulation domain, the main grid has $100\times100$ cells in horizontal direction for case S3 and $50\times50$ cells for case S4. All of other model setups (including vertical grid size and number) and initial conditions are same for these cases.

  The simulated $NO_2$ profiles for two additional grid resolutions are plotted in Fig. B1, which shows similar results as the

reference case (Fig. 8) but with slightly different details. In one hand, the space-varying wind fields for the different grid-resolution simulations are slightly different from the value in the reference case (case 1). As mentioned in last section, the parameters of the plume segments, such as location, size and speed, will be affected for the cases with different horizontal grid resolutions. In another hand, the plume mass will be added into the main grid cell that the plume locates, and then it will induce a relatively higher local concentration for the finer grid and lower concentration for the coarse grid, compared to the

reference case. Therefore, the integration of the sub-grid plume model with the main Eulerian model will be affected to result in different $NO_2$ concentrations for the different grid setups. However, compared to the coarse grid (case S4), when the grid size is reduced, the simulation results for cases 1 and S3 are much closer to show mesh independence, as presented in Figs. 8 and B1.

  Similar study was conducted by changing the vertical grid size (dz) as well. In this study, a smaller dz (especially in lower

height) was applied in the simulation (case S5: 20 vertical layers), compared to the reference case (case1: 13 vertical layers). All other setups and conditions are same for the two cases. Similar conclusions are obtained that the simulated $NO_2$ profiles are very similar with only slightly different details for the two different setups, as shown in Fig. 8 and B1. The parameter study suggests that the MPS model works well for the shipping emission dispersion modelling.

*Author contributions.* KP developed the model code, performed the simulations, processed and evaluated the results. EM discussed the model

development and provided suggestions for the model evaluations. KP prepared the manuscript with the contributions from all co-authors.

*Competing interests.* The authors declare that they have no conflict of interest.

*Acknowledgements.* This study is funded by the National Research Foundation, Prime Minister's Office, Singapore under its CREATE programme. The authors appreciate the support for EPISODE installation from Dr. Matthias Karl in Helmholtz-Zentrum Geesthacht, Department of Chemistry Transport Modelling, Geesthacht, Germany. KP also thanks Yichen Zong and Arkadiusz Chadzynski for their suggestions and

help on obtaining input data for simulations.







(a) At $t = 60$ min      (b) At $t = 120$ min      (c) At $t = 180$ min

**Figure B1.** Instantaneous NO$_2$ concentrations near ground by using different grid resolutions. $1^{st}$ row: dx=dy=700m (case S3); $2^{nd}$ row: dx=dy=1400m (case S4); $3^{rd}$ row: small dz (case S5).





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
