# Peer review of "Development of a moving point source model for shipping emission dispersion modelling in EPISODE-CityChem v1.3"

_Geoscientific Model Development, 2021_

## Author Comment (AC1)

**General comment**

The paper presents a new modelling approach to take into account movements of ships in simulation of transport and dispersal of atmospheric pollutants. Some specific cases have been simulated and compared to other modelling approaches such as fixed emission points and line source models. The paper is interesting, suitable for the Journal and could be of interest also for future studies. However, there are several aspects that could be made more clear and also an over-interpretation of results because I believe that it is not very clear that this new modelling approach is performing effectively better compared to other approaches at least when several ships are considered (see my specific comments). In conclusion, I believe that the paper should have a carful revision before publication.

Response:
The authors appreciate the many constructive comments and suggestions made by the reviewer. We have revised the manuscript and conducted some additional simulations to better present our results and conclusions. More details about the responses to the reviewer's detailed comments are listed below (the reviewer's comments are in "blue" and the responses are in "black").

**Specific comments**

1)  One aspect that raised my curiosity is why SO2 has not been considered in this work. Shipping-related SO2 is quite important and the most recent international policy from IMO enforce the use of low-sulphur content fuel that will have a strong impact on the emission of SO2 (in addition to particulate matter). In addition, it is mentioned that simulations were performed also for PM2.5 but I only see NO2 in the results.

Response:
In our simulation, actually $SO_2$, $NO_2$, $PM_{2.5}$ and some other important species are predicted. However, the distributions of all species are very similar when using the same emission model (such as MPS model), and only the concentration values are different. Since the purpose of this paper is to demonstrate the development of the moving point source (MPS) model and its difference from the line source (LS) and fixed point source (FPS) models, only $NO_2$ is presented in the simplified cases (section 3.1 and 3.2), as the simulated results for all species show the same conclusions.

In addition, in the real case study (section 3.3), $NO_2$ and $PM_{2.5}$ curves showing comparisons of simulated results with measured data are presented as Figs. 17a - 17d, and new figures (Fig. 18 in the revised paper) showing the $NO_2$ and $PM_{2.5}$ concentration differences predicted by the LS and MPS models during the entire simulation period are added as well. In the revised paper, a new appendix (Appendix D) has been added to explain this and also shows the similarity of the predicted $NO_2$, $SO_2$ and $PM_{2.5}$ in the simulations.

2)  In several parts of the paper there is a confusion between "emissions" and concentrations". In line 154, 162, 202-205 I believe that authors are mentioning actually concentrations rather than emissions. Lines 171-175. These limitations are effectively necessary? What are the reasons behind these choices?

Response:
Yes, we are mentioning about concentrations. In the revised paper, we have changed the words to "concentrations" (line 154, 162 and 202-205 in original paper). As shown in line 171-175, a simplified case is setup first by only keeping the moving ships, in order to better demonstrate the feature of the MPS

model and the differences when using MPS model to simulate the emission dispersion profiles generated from moving ships, compared to those predicted by using LS and FPS models. If too complicated situations (such as including ships at berthed and using different wind direction) are included or simulated, it is not easy to identify the differences of MPS model with LS and FPS models. Therefore, the paper used these choices as shown in line 171-175 in the simplified cases. New materials have been added to explain this in the revised paper (pg. 9, lines 180-181).

3) The simulations cover a very limited period (only a few hours). Will the results of the comparison among the different models similar if larger time span are used for simulations? Often daily or seasonal averages are used to investigate the impacts of specific sources to air quality. I ask this because, it seems that when a large number of ships are included, the differences of moving ship model and line emission model become negligible.

Response:
The authors thank the reviewer for the question. The results by using the MPS model and other two models (FPS and LS models) are different depends on the locations. Based on the figures in the simplified study (sections 3.1 and 3.2), it indicates that the instantaneous results predicted by the three different models are quite different, and the hourly-averaged results are still different (especially the concentration values) although the emission species distributes in a similar shape when using the MPS and LS models.

In addition, in the revised paper, new figures (Fig. 18) showing the differences of overall averages (6 hours) predicted by using the MPS and LS models for the real-case study are added. Based on the new figures, it shows that the predicted emission concentrations are very different in the locations close to the ships (or ship routes), due to the different treatments of the two models on emission sources, and the concentration differences become smaller in the locations far away from the ships, as the emissions are diluted and deposited. New materials have been added in the revised paper to explain this (pg. 20 and 21).

4) What are the emission conditions such as vertical exit velocity, height of emissions, buoyancy and so on? Are the same conditions used for each ship or a difference has been done according to the typology of ship (for example cargo, cruise, ferry and so on). Could author comment if the uncertainty arising from the assumption made on emissions is smaller or comparable with the differences observed among the different models?

Response:
In this study, the vertical exit velocity for all ships is assumed as 20m/s, and the chimney heights (where emission is emitted) are different for different ships (30m for large ships (such as cargo, container) and 10m for small ships (such as pleasure, fishing)). The buoyancy was calculated based on Briggs's algorithms [Briggs 1969, 1971 and 1975], which consider the different atmospheric stability conditions (such as neutral-unstable and stable conditions). In addition, stack downwash and plume penetration are estimated as well in EPISODE package. The emission conditions have been mentioned in pg. 4 (line 112-114), pg. 9 (line 193-196) and new Appendix C (pg. 26) in the revised paper.

The authors also conducted sensitivity studies that consider different emission heights which are affected by exit velocity, chimney height and buoyancy, and the simulated results showed that the uncertainty arising from the assumptions made on emissions is very small, which is much smaller than the differences among different models. The results of the sensitivity studies can be found in Appendix C of the revised paper.

5) Figure 7. It is actually difficult to compare results because the colour scale is very different. It should have been better to have all figures with the same colour scale. The same for Figs. 13, 14, 15

Response:
New figures with same color scale have been added to the revised paper for Figs. 7, 13-15.

6) Figures 16 and 17 are based on only a few points so that the correct line joining the points is a straight line rather than a "non-estimated" curve.

Response:
New figures with straight line connecting points have been added to the revised paper for Figs. 16 and 17.

7) Lines 349-360. The better performance of MPS is not really visible. Results of MPS and line source in Figure 17 are essentially the same with negligible differences especially if compared to the uncertainty rising from assumption on emissions. Therefore this part of the discussion should be revised.

Response:
The authors agree with the reviewer that the MPS and LS models perform similar when compared to the measured data in the observation stations, based on the results (Fig. 17) in this study. This has been clearly mentioned in the discussion part of the revised paper. In addition, new figures (Fig. 18) showing the differences of the 2D emission concentrations predicted by the MPS and LS models have been added to the revised paper as well. Based on the new figures, it indicates that the predicted concentrations by the two models are quite different in the locations close to the ships while the differences become smaller in the locations far away. The new figures and other results (those in the simplified simulations) in the paper suggest that the new MPS model is a more realistic representation of the emission source, allows for greater granulation of the emissions and is expected to be reasonably accurate, and hence it should be a valuable alternative for the environmental researchers to investigate and evaluate the dispersion of emissions generated by the moving ships. In the revised paper, the paragraph in the discussion (pg. 20 and 21) has been modified to better describe the results and important conclusions.

8) The same thing for the conclusions (line 370-373), it is possible that, for very short calculation periods and with a very limited number of ships, MPS could furnish more realistic results. However, this aspect is not really demonstrated by comparison with measurements. Regarding the conclusions (lines 383-387) of "real-world" simulation, I believe that it is not true that MPS furnish better results than line source model. The differences found among the two approach are negligible. This should be clearly stated in the conclusion of the paper.

Response:
The authors thank the reviewers for the comments. It is true that the differences predicted by the MPS and LS models are negligible, compared to the measured data (as shown in Fig. 17) when a large number of ships are included, and this has been clearly mentioned in the conclusion part. However, as mentioned in responses for Q7), the results in the paper also suggest that the new MPS model is a more realistic representation of the emission source, allows for greater granulation of the emissions and is expected to be reasonably accurate, and hence it should be a valuable option for the environmental researchers to investigate and evaluate the dispersion of emissions generated by the moving ships. In the revised paper, the conclusion part has been modified to better demonstrate the important results and conclusions about the new MPS model.

---

## Author Comment (AC2)

The paper presents the development of a moving point source (MPS) model and compares it with two other common methods for evaluating the local dispersion of ship emissions, namely the line source model and the fixed point source model. The MPS was implemented as sub-grid module in the urban chemistry transport model EPISODE-CityChem to study the impact of ship emissions in numerical experiments with one and several ships, as well as in a real-world scenario in Singapore. The simulations are carefully done and the results properly discussed. MPS has a great potential when used together with AIS ship position data for real-time simulation of pollutant dispersion from ship emissions. The moving point source model is a valuable addition to the EPISODE model for assessing impacts of ship emissions on air concentrations and human health on local and city scale, allowing for more details on the spatial and temporal distribution. The comparison between MPS and the line source model reveals that differences between the two methods are more obvious for instantaneous concentrations than for longer averaging times (1 h), because the dispersion of single plumes released at different points along the same trajectory (line) becomes more homogeneous in space when longer time scales are considered. The real case did not reveal the clear benefits from using MPS, probably because the period of five hours was too short to cover different weather conditions, changes in the boundary layer structure or day/night variation. My suggestion is to include one more case that studies the role of different atmospheric stability conditions on the ship impact from the three emission models. Overall, I am in favor of publishing the paper after my specific comments below are sufficiently addressed.

Response:
The authors thank the reviewer for the valuable comments and suggestions. We have modified the manuscript, conducted more comparisons and run additional simulations to better present our results and conclusions. More details about the responses to the reviewer's comments are listed below (the reviewer's comments are in "blue" and the responses are in "black").

**Specific Comments:**

1.) The abstract should better reflect the results from the study and give quantitative information about the evaluation of the performance of MPS. In particular, the larger discrepancies between the emission source models for instantaneous than for averaged concentrations should be stated more clearly.

Response:
The authors thank the reviewer for this suggestion. In the revised paper, the abstract has been modified to better demonstrate the important results and conclusions from the simulations conducted in this paper.

2.) Is there any specific treatment when the plumes released from different virtual points during a simulation hour are crossing or overlapping each other? There should be some assumption about merging of the plume masses or other interaction between the individual plumes.

Response:
For one ship, an individual plume is emitted from each virtual point during each timestep, and it is then treated by using the Gaussian segmented plume model (SEGPLU) available in EPISODE. In each timestep, the parameters (such as size, location and so on) of each individual plume will be calculated, and then all mass of one plume will be merged into the Eulerian cells for further 3D convection-diffusion calculation, when its size grows to a predefined value ($\sigma y/dy=4$ or $\sigma z/dz=4$). The contributions of all existing plumes on sub-grid will then be calculated in each timestep as well. By using these methods, the contributions of

the plumes are estimated, even when they are crossing or overlapping each other. New material has been added to the revised paper in Pg. 5.

3.) Figure 4: what is causing the structures in the wind field?

Response:
In EPISODE, the build-in meteorological pre-processor code, MCWIND, is used to calculate the wind field in Singapore. In the calculation, the estimated wind speed and direction will be adjusted based on the local topographical conditions, so that the wind speed and direction are not always same in different locations as shown in Fig. 4. This has been mentioned in section "2.3.1 Simplified study" (pg. 7, line 173-174).

4.) In general, there is too little information about the plume rise algorithm. How is plume rise handled in the LS model?

Response:
As a default method used in EPISODE, the plume rise (due to buoyancy or momentum) was calculated based on Briggs's algorithms [Briggs 1969, 1971 and 1975], which consider the different atmospheric stability conditions (such as neutral-unstable and stable conditions). In the revised paper, new materials have been added (pg. 4, line 111-113) to mention this information. In addition, the default LS model in EPISODE-CityChem was designed for estimating the emission dispersion generated from cars, which emit emissions at 1m above ground. In our study, the LS model was modified to consider the plume rise estimated by Briggs algorithms as well.

5.) What is assumed about the ship building height and width, since they can influence the plume rise?

Response:
The ship building height (BH) and width (BW) are assumed to have different values for different ships. The building height is assumed to be 5 m below the ship chimney heights. Specifically, BH is 25 m and BW is 20 m for large ships (such as cargo, container), while BH and BW are both 5 m for small ships (such as pleasure, fishing). New materials have been added to the revised paper (pg. 9, line 193-195) to describe the building height and width for different ships.

In addition, an additional simulation was conducted to use different BH and BW values (20m BH and 15m BW for all kinds of ships) as well, and very similar results are found when compared to the ship building setups used in this study. The results for the sensitivity study can be found in the new Appendix C in the revised paper.

6.) Real case study, section 3.3: a longer simulation period could reveal discrepancies between MPS and the LS model. The differences between the two models are very small and based on the real case it is currently not possible to conclude that MPS performs better. It is suggested to show the average 2-D fields for the observation period, to analyze where the largest discrepancy between the models occur and to look at a time series in the place of largest impact from ships. Differences in hourly average NO2 concentrations can be noticed, for example when looking at the 2-D plots in Fig. 15 at 180 min simulation time, over the eastern part of the city. It may be considered to show 2-D maps of concentration differences, to make such details clearer.

Response:
The authors thank the reviewer for the suggestions. New figures (Fig. 18) showing the differences of overall concentration averages from MPS and LS simulations have been added in the revised paper. The figures (2D maps) showed that the biggest differences between two models are in the locations close to the ships

especially where large number of ships are located, while the differences are reduced to very small in the locations far away from the ships (such as the inland of Singapore, where observation stations are), due to the emission dilution and deposition. Additional figures (Fig. 18c and 18d) showing the time series of the emission concentrations at the location where the big differences exist were added to the revised paper as well, which once again support that the predicted results by the two models are quite different near the ships.

We also tried to conduct additional simulations to include different weather conditions and day/night variations, however, the results are still same to those in Figs. 17 and 18 that the MPS and LS results are still similar in the location of observation stations while large differences exist in the locations close to ships. Therefore, it is very hard to get the conclusions that the new MPS model will predict better than the LS model when compared to the measured data. However, the comparison between the MPS model and the measured data (Fig. 17) indicates that the new MPS model could reasonably predict the dispersion of shipping emissions. In addition, the simplified results and the 2D plots in the real cases also showed that the new MPS model is a more realistic representation of the emission source and allows for greater granulation of the emissions. Hence, the MPS model could be a useful addition or alternative for the environmental researcher to evaluate the dispersion of emissions generated by the moving ships. In the revised paper, the discussion in "3.3 Real case study" and the conclusion parts have been modified to better describe the results and conclusions drawn from this study.

**Technical Corrections**

7.) 10 line 20: emission and concentration are confused here.

Response:
The sentences have been modified in the revised paper to make it clearer.

8.) 18 line 328: mention that locations of observation stations are shown in Fig. 6.

Response:
In the revised paper, the sentence has been modified to indicate that the locations of the observation stations are shown in Fig. 6.

9.) 18 line 330: replace "may contributed from following aspects" by "may be attributed to the following aspects".

Response:
In the revised paper, the sentence has been modified as "may be attributed to the following aspects."

10.) 19 line 339: replace "very" by "vary".

Response:
In the revised paper, "very" has been replaced by "vary".

11.) Figure 15: MPS and LS model 2-D plots ought to have the same color scale to allow for a quantitative comparison. This also concerns the other 2-D plots, like Fig. 13 and Fig. 14.

Response:
In the revised paper, new figures with same color scale have been added to the revised paper for Figs. 13-15.

---

## Author Response (AR2)

**Responses to reviewers' comments on gmd-2021-47**

The authors appreciate the many constructive comments and suggestions made by the reviewer. The paragraphs below address each comment in turn.

**Anonymous Referee #2**

The revised manuscript is much more convincing in demonstrating that the moving point source model has significant benefits compared to the line source model when dealing with the dispersion of ship emissions. I have only a few remaining remarks that should be handled before publication in Geoscientific Model Development:

1. In the Introduction, the difference between the two main types of atmospheric models (Lagrangian and Eulerian models) and the Gaussian plume models should be described more clearly. I suggest to rearrange the text on page 2, by first introducing the Gaussian models, their application to ship emissions, and their limitations. After that, introduce the Eulerian and Lagrangian model types, their differences, as well as their application to ship emission dispersion.

Response:
The authors thank the reviewer for the suggestions. In the revised paper, the introduction has been modified to better describe the different types of dispersion models (pg 2).

2. In section 2.2, explain in more detail how the additional parameters are derived from AIS ship position data.

Response:
The authors thank the reviewer for the question. Three additional parameters (namely turning angle, start time and stop time) are defined to provide the options to customize the ship movement, if the exact values of the additional parameters for one ship can be obtained from either the ship traffic website or other databases, so that they can make the MPS model more flexible and realistic.

However, in our study, we didn't get the values of the additional parameters for each ship, due to the lack of such data. For this reason, in each simulation hour, the start time ($t_{S1}$) and stop time ($t_{S2}$) of all moving ships are assumed to be 0s and 3600s respectively, meaning that the ships keep moving for the entire hour. In addition, the turning angle ($\theta$) is set as 0° in this study, meaning that the ship is assumed to move straightly. In the revised paper, new materials have been added in pg. 5 (line 137-138) and pg. 7 (line162-164).

3. Figure 18, in the 2-D maps of differences in figure parts a) and b) the color bar title should be "delta NO2" and "delta PM2.5".

Response:
In the revised paper, the color bar title has been modified as "$\Delta NO_2$" and "$\Delta PM_{2.5}$" for Fig. 18(a) and 18(b).